# Anti-Mullerian Hormone-Based Phenotyping Identifies Subgroups of Women with Polycystic Ovary Syndrome with Differing Clinical and Biochemical Characteristics

**DOI:** 10.3390/diagnostics13030500

**Published:** 2023-01-30

**Authors:** Minhthao Thi Nguyen, Sridevi Krishnan, Sonal V. Phatak, Sidika E. Karakas

**Affiliations:** 1Department of Internal Medicine, Division of Endocrinology, Diabetes and Metabolism, University of California Davis School of Medicine, Sacramento, CA 95817, USA; 2Department of Pediatrics, Glycobiology Research and Training Center, University of California San Diego School of Medicine, La Jolla, CA 92037, USA

**Keywords:** anti-Mullerian hormone (AMH), polycystic ovary syndrome (PCOS), phenotyping, subtypes of PCOS

## Abstract

Even though polycystic ovary syndrome (PCOS) was originally defined as “amenorrhea associated with bilateral polycystic ovaries”, women without PCO morphology can be included in this diagnosis. This may contribute to the clinical heterogeneity seen in PCOS. Serum anti-Mullerian hormone (AMH) correlates with the number of ovarian cysts. We investigated whether phenotyping based on serum AMH can distinguish subgroups of PCOS with different clinical and biochemical characteristics. The electronic medical records of 108 women with PCOS (Rotterdam criteria) were reviewed. The serum AMH value correlated inversely (0.03 < *p* < 0.0001) with age, weight, and BMI values and directly with serum total testosterone (T), free T, and bioavailable T values. When divided into quartiles based on serum AMH values, the women in the highest quartile (AMH: 18.5 ± 9.9 ng/mL; n = 27) had lower BMI (29.4 ± 6.9 vs. 34.0 ± 10.6–36.7 ± 7.2 kg/m^2^) but higher total T (51.3 ± 27.2 vs. 26.5 ± 10.4–35.1 ± 16.3 ng/dL), free T (7.7 ± 6.0 vs. 4.4 ± 2.3–5.7 ± 3.2 ng/dL), and bioavailable T (22.1 ± 17.0 vs. 12.2 ± 6.6–16.5 ± 8.7 ng/dL) values. The combination of high AMH and high testosterone values may point to the ovaries and reproductive etiology for PCOS in this subgroup. Thus, AMH-based phenotyping may provide a practical and cost-effective tool to explore the heterogeneity in PCOS.

## 1. Introduction

Polycystic ovary syndrome (PCOS) was defined originally by Drs. Stein and Levinthal as “amenorrhea associated with bilateral polycystic ovaries” in 1935 [1]. Since then, polycystic ovary (PCO) morphology has remained as one of the cardinal but not mandatory criteria for the diagnosis for PCOS. Subsequently, the definition evolved to emphasize different aspects of the syndrome, such as androgen excess (biochemical or clinical) and amenorrhea–oligomenorrhea–anovulation, in addition to the polycystic ovary morphology [2]. The National Institutes of Health (NIH) criteria (1990) required hyperandrogenism and oligomenorrhea as mandatory criteria, while PCO was not required for diagnosis [3]. The Rotterdam criteria (2003) gave equal diagnostic value to all three criteria, with the presence of any two being adequate for a diagnosis. The Rotterdam definition of PCOS was criticized because women without hyperandrogenism could still be diagnosed as having PCOS [4]. Subsequently, in 2006 the Androgen Excess PCOS Society (AE-PCOS) published another set of diagnostic criteria, making androgen excess a mandatory requirement and having either oligo-ovulation–anovulation or PCO morphology as the secondary requirement for the diagnosis of PCOS [5]. Women diagnosed as having PCOS based on these different diagnostic criteria display four different clinical phenotypes: phenotype A: hyperandrogenism + anovulation + PCO; phenotype B: hyperandrogenism + anovulation; phenotype C: hyperandrogenism + PCO; phenotype D: anovulation + PCO [2].

Ultrasound has historically been the primary method used to assess PCO morphology. The diagnostic criteria include a follicular size of less than 9 mm, an ovarian volume of more than 10 mL, and an increased number of follicles. The follicle number per ovary (FNPO) requirement has increased from 10 to 12 to 23 along with the advances in ultrasound technology [6,7]. However, determining the FNPO is dependent on the technician and the ultrasound equipment, which can limit the standardization and reproducibility of the results at different medical centers. The use of ultrasound for a PCOS diagnosis is not recommended until eight years after menarche [7]. Transabdominal ultrasound can be used to assess the ovarian volume; however, it is not as sensitive as the transvaginal approach to assess the number of cysts, and may deter patients due to its invasive nature. 

Studies conducted over the last two decades demonstrated that serum anti-Mullerian hormone (AMH) provides a practical, cost-effective biomarker to assess PCO morphology, with the serum AMH levels correlating with the follicle count determined by ultrasound [8,9,10,11,12,13]. Anti-Mullerian hormone is secreted by the granulosa cells, with the expression starting in the primordial follicles and reaching its peak in the preantral and small antral follicles [14]. As the follicles increase in size and become FSH-dependent, the AMH production decreases [15]. The total number of ovarian follicles is increased in women with PCOS. This is especially true for the pre-antral and antral follicles, which produce large amounts of AMH [16]. In addition, larger follicles continue to express and secrete higher amounts of AMH [17]. The serum AMH levels are usually 2–3 times higher in women with PCOS than control women. Several studies demonstrated that serum AMH levels above 5 ng/mL (35.7 pmol/L) can help to diagnose PCOS [12,18], even though it is not as sensitive as transvaginal ultrasound [19]. Because of technical concerns related to the assay, serum AMH testing has not yet been accepted as a replacement for ultrasound [20].

In addition to providing a valuable biomarker for the PCO morphology and follicular reserve, AMH has a significant role in reproductive function. Studies in mice indicated that AMH interferes with the stimulation of granulosa cells via FSH and estrogen production in the ovaries [21]. Anti-Mullerian hormone regulates GnRH, LH, and FSH production in the hypothalamus and pituitary gland [22,23]. Consistent with these experimental findings, PCOS patients with high serum AMH concentrations are resistant to gonadotropins and require higher doses for ovulation induction [24].

Since ovarian cysts are a critical component of PCOS, high AMH concentrations reflect PCO morphology, and AMH may regulate both the central secretion and ovarian action of gonadotropins, we hypothesized that women with PCOS with higher serum AMH concentrations may differ phenotypically from those with lower serum AMH levels. Emerging reports based on genetic studies have identified three subtypes of PCOS: reproductive, metabolic, and indeterminant [25,26]. Since the serum AMH concentration was not measured in these studies, it is not known whether these subtypes differ in their serum AMH levels. We thought that women with PCOS with high serum AMH levels may have a primary pathology residing in their ovaries, which may be similar to the reproductive subtype. On the other hand, women with PCOS with low serum AMH levels may not have ovarian cysts; their primary pathology may not reside in the ovaries, but they may develop PCOS because of obesity and insulin resistance, which may be similar to the metabolic subtype. So far, our results show that women with PCOS who are at the high end of the serum AMH spectrum have high levels of ovarian androgens but they are leaner. In contrast, the women at the low end of the serum AMH spectrum have lower ovarian androgen concentrations but they are more obese. Thus, AMH-based phenotyping is able to discriminate women with PCOS with different anthropometric and hormonal characteristics. Our findings suggest that the measurement of serum AMH is a promising tool for understanding the clinical heterogeneity seen in PCOS.

## 2. Materials and Methods

The study was approved by the Institutional Review Board of the University of California, Davis (approval code: 1841327-1; approval date: 21 March 2022).

### 2.1. Subjects

The EPIC EMR was used to retrospectively review the charts of the patients who were referred to the PCOS clinic at the UC Davis Medical Center for a presumed diagnosis of PCOS. Those who fulfilled the Rotterdam criteria (presence of 2 out of 3 findings of hyperandrogenism, anovulation–oligomenorrhea–amenorrhea, PCO morphology of the ovaries) were included. The exclusion criteria were the use of oral contraceptives during the time of the blood draw, and other diagnoses mimicking PCOS (i.e., congenital adrenal hyperplasia, prolactinoma, premature ovarian failure).

The clinical data included weight, height, body mass index (BMI), and systolic and diastolic blood pressure (SBP and DBP) values. The laboratory data included AMH, testosterone (T), SHBG, bio-available T, free T, dehydroepiandrostenedione-sulfate (DHEA-S), 17-hydroxyprogesterone (17-OHP), pituitary hormone (LH, FSH, prolactin), thyroid stimulating hormone (TSH), and metabolic variables (HgA1C, total-cholesterol (C), HDL-C, LDL-C, and triglyceride).

The patients’ histories were reviewed and subjective complaints of absence for periods of more than six months (amenorrhea), infrequent periods of more than 35 days apart or less than nine periods per year (oligomenorrhea), frequent excess bleeding (menorrhagia), excess facial or body hair (hirsutism), weight gain and obesity, and use of metformin and levothyroxine were recorded.

### 2.2. Laboratory Methodology

Blood samples were obtained during the 3–5 days of the menstrual cycle in patients who had menstrual periods. In amenorrheic–oligomenorrheic patients, the blood samples were obtained randomly. The anti-Mullerian hormone concentrations were measured by the ARUP Laboratory (Salt Lake City, UT, USA) using a quantitative enzyme-linked immunosorbent assay. The testosterone (mass spectrometry), sex-hormone-binding globulin (quantitative electrochemiluminescent immunoassay), bioavailable testosterone, and free testosterone (calculated from testosterone and SHBG or albumin values) concentrations were also measured and reported by the ARUP Laboratory (Salt Lake City, UT, USA). All other laboratory assays were carried out by clinical chemistry laboratories of the University California Davis Medical Center (Sacramento, CA, USA).

### 2.3. Statistics

The descriptive statistics are presented as means ± standard deviations (SDs). The subjective data were evaluated as frequency values. Spearman correlations were used to evaluate the relationships among clinical, biochemical, and subjective variables. To identify the women with PCOS with the highest serum AMH concentrations, the subjects were divided into quartiles, based on the 25th, 50th, 75th, and 100th percentile concentrations of AMH. The serum AMH ranges for the 1st quartile (minimum: 0.1; maximum: 4.0 ng/mL (mean ± SD: 2.1 ± 1.1 ng/mL)), 2nd quartile (minimum: 4.1; maximum 6.2 ng/mL (mean ± SD: 5.1 ± 0.6 ng/mL)), 3rd quartile (minimum: 6.6 ng/mL; maximum 10.3 ng/mL (mean ± SD: 8.0 ± 1.1 ng/mL)), and 4th quartile (minimum 10.5 ng/mL; maximum 52.5 ng/mL (mean ± SD: 18.5 ± 9.9 ng/mL)) were obtained. As a result of this quantile binning distribution, quartiles 1 and 4 included 27 patients each, while quartiles 2 and 3 included 25 and 29 patients, respectively.

All patients had the anthropometric data and essential laboratory data necessary for a diagnosis of PCOS (testosterone, bioavailable T, free T, SHBG, and AMH), and the quartiles were determined based on AMH. However, there were missing data points for the pituitary hormones (LH, FSH, prolactin), TSH, adrenal androgens (17OHP, DHEAS), and metabolic parameters (lipids, HgBA1). Therefore, a statistical analysis was performed to compare all available complete cases for each variable.

All data were evaluated for normality using Q-Q plots and Shapiro–Wilk tests. Since the data were not normally distributed, in order to evaluate differences between four quartiles at once, ANOVAs were conducted on transformed data. The data were transformed using Johnson distribution and sine–arcsine transformations as appropriate (with the exception of FSH, which was normally distributed and did not need transforming). Clinical, biochemical, and anthropometric variables (continuous data) were compared across quartiles using ANOVAs followed by Tukey’s pairwise comparisons when the ANOVA was significant. The categorical data were analyzed using chi square likelihood ratios for tests of independence. The analyses were performed using JMP Pro version 16.2.0 (SAS Institute, Cary, NC, USA).

## 3. Results

### 3.1. Clinical Characteristics of the Women with PCOS

The clinical, biochemical, and subjective characteristics of the 108 women with PCOS are summarized in Table 1. Subjectively, 62% complained of oligomenorrhea, 34% of amenorrhea, 51% of facial hair, 19% of scalp hair loss, 42% of obesity, and 41% of weight gain. The rates of subjective complaints did not differ significantly among the quartiles (Table 2). Five patients (5.2%) were taking levothyroxine. Thirty-four percent of the patients used metformin (500–1500 mg/d; 19% in quartile 1, 40% in quartile 2, 24% in quartile 3, and 7% in quartile 4).

### 3.2. Relationships between Anti-Mullerian Hormone and Clinical and Biochemical Characteristics of Women with PCOS

The significant correlations among the clinical and biochemical variables are summarized in Table 3. The serum AMH value correlated inversely with age, weight, and BMI values and directly with serum total T, free T, and bioavailable T values. The body mass index correlated inversely with DHEA-S but directly with HgBA1. Sex-hormone-binding globulin correlated inversely with BMI, HgBA1, bioavailable T, and free T values and with subjective complaints of obesity and weight gain.

### 3.3. Clinical and Biochemical Characteristics of Women with PCOS in Different Serum AMH Quartiles

The mentioned values are shown in Table 2 and Figure 1. The quartiles did not differ significantly with age. Overall significant differences were seen in weight, BMI, total T, bioavailable T, and LH values. The free T, 17OHP, and HDL-C values showed borderline differences (*p* < 0.1). Women in quartiles 1 and 2 were significantly more obese than those in quartile 4 (highest AMH concentrations). The highest AMH quartile had higher total T concentrations as compared to the other quartiles. The luteinizing hormone concentration in the highest AMH quartile was higher than that in quartile 2 but did not differ significantly from the other quartiles.

## 4. Discussion

The anti-Mullerian hormone-based phenotyping showed that the women with PCOS in the highest AMH quartile also had higher total T and bioavailable T concentrations. These findings agree with the recent report by Simons et al. who showed that serum AMH is associated with total T and androstenedione (A4) and antral follicle count is associated with total T, free T, and A4 in women with PCOS [27]. Feldman et al. reported direct correlations between serum AMH and ovarian androgens in PCOS as well [28]. Their study focused on cardiometabolic risk and included 252 obese women with PCOS. They found that serum AMH correlated directly with healthy metabolic risk markers such as lower BMI, high HDL-C, high SHBG, and low HOMA concentrations.

We observed an inverse correlation between weight and serum AMH. Our highest AMH quartile had lower body weight values as compared to the women in quartiles 1–3. Several other studies showed similar relationships. Specifically, obesity was found to be associated with lower AMH levels in women with or without PCOS [29]. The studies by Carmina et al. also linked PCO morphology to leanness. Their study investigated the prevalence of obesity in different phenotypes of PCOS [30]. Out of 247 women with PCOS, 199 belonged to phenotypes that included PCO morphology (A, C, and D); 48 patients belonged to phenotype B, which does not require PCO for a diagnosis. Women belonging to the phenotypes including PCO morphology were leaner. The percentages of lean vs. obese women in the phenotypes requiring PCO morphology were as follows: phenotype A: 43% lean vs. 33% obese; phenotype C: 64% lean vs. 11% obese; phenotype D: 43% lean vs. 14% obese. In contrast, phenotype B, which did not include the PCO morphology, showed a reversal of this ratio, with 25% lean vs. 54% obese. 

The positive relationship between serum AMH concentrations and leanness may raise a question about the cause-and-effect relationship: Is it possible that the high AMH values seen in our highest AMH quartile were due to their lower body weight? To answer this question, we stratified the patients based on their BMI values (data not shown). The patients in the lowest BMI quartile had a BMI value of 23.5 ± 2.7 kg/m^2^ (n = 27). This was lower than the BMI of the highest AMH quartile patients (BMI: 29.4 ± 6.9 kg/m^2^, n = 27). If being lean was the driving cause of having high AMH levels, the lowest BMI group should have had even higher serum AMH levels, but they did not. They had a lower serum AMH concentration of 12.0 ± 10.8 ng/mL, lower testosterone concentration of 40.8 ± 18.0 ng/mL, lower free T concentration of 4.8 ± 2.8 ng/mL, higher SHBG concentration of 64.9 ± 26 nmol/L, and a comparable HgBA1 concentration of 5.4 ± 0.5%. These findings indicated that having a lower weight was not the primary factor driving the high serum AMH and high ovarian androgen concentrations seen in the highest AMH quartile. 

A high serum AMH concentration points to increased cysts and follicles, while a high testosterone concentration indicates increased theca cell activity in the ovarian stroma [2]. Taken together, these point to the ovaries as the primary culprit in PCOS in the highest AMH quartile. Rosenfield and Ehrmann proposed a crosstalk between the theca cells and ovarian follicles. In PCOS, the theca cells are hyperresponsive to LH and insulin [31]. This results in the increased production of androgens such as 17OHP and testosterone. Increased testosterone leads to preferential increases in smaller follicles secreting AMH. High testosterone along with high AMH concentrations decrease the responsiveness of the granulosa cells to FSH, causing anovulation and leading to PCO [21,32].

Based on GAWAS studies, Dapas et al. described three different subtypes of PCOS [26]. In their report, 23% of patients displayed primarily the reproductive characteristics of PCOS, while 37% showed primarily the metabolic characteristics and the remaining 40% was indeterminant. Compared to the metabolic subtype, the reproductive subtype patients were leaner, less insulin-resistant, and had higher LH and FSH concentrations. The reproductive and metabolic subtypes had similar testosterone concentrations, while serum AMH concentrations were not measured. 

Our highest AMH quartile patients showed some similarities but also differences when compared to the reproductive subtype of PCOS described by Dapas et al. The similarities were that our highest AMH quartile was significantly leaner than the rest of the study population and showed some elevation of LH as compared to one of the other quartiles. The major difference between our highest AMH quartile and the reproductive subtype was that our highest AMH quartile had significantly elevated ovarian androgen levels, whereas the reproductive subtype did not have elevated testosterone levels. The reproductive subtype described by Dapas et al. was more insulin-sensitive as compared to the other subtypes. Insulin resistance could not be assessed in our study because of the lack of necessary data (i.e., fasting insulin, glucose, or HOMA levels). However, our highest AMH quartile tended to have lower HgBA1 concentrations as compared to the other quartiles (5.3% vs. 5.6%), despite having a lower metformin use rate (7%). In the other quartiles, 19% to 40% of the patients were taking metformin.

An interesting observation was that the European GWAS data associated the reproductive subtype to a genetic locus in the type-I AMH receptor bone morphogenetic protein receptor type 1B (BMPR1B). which relates to the follicular reserve [33]. It will be important to investigate whether the reproductive subtype has alterations in serum AMH. The European GWAS study associated the reproductive subtype also to another genetic locus involving an estrogen receptor co-activator PRDM2, which is expressed in the pituitary gland and the ovaries and acts in the development of granulosa cells [34]. These findings suggest that reproductive subtype of PCOS has a genetic background [25]. A recent report by Zhang et al. also associated the reproductive, obese, and insulin-resistant subtypes of PCOS to different genetic traits [35]. 

We entertained the possibility that our lowest AMH quartile (quartile 1) may share similarities with the metabolic subtype described by Dapas and Feldman [26,28]. Our lowest AMH quartile patients (quartile 1) had higher BMI values than the quartile 3 and quartile 4 patients, but their values overlapped with those of the 2nd quartile. This may be because in the studies by Dapas, the metabolic subtype comprised 37% of the PCOS population, which extended beyond one quartile. Our lowest AMH quartile differed from the metabolic subtype in terms of serum testosterone levels, as the lowest AMH quartile had lower testosterone levels as compared to the highest AMH quartile (*p* = 0.0215), whereas the reproductive and metabolic subtypes defined by the European GAWAS study did not differ in their testosterone levels. 

Our study demonstrated that anti-Mullerian hormone-based phenotyping provided several novel and significant findings. The patients at the highest end of the serum AMH spectrum displayed significantly different clinical and biochemical features. Those in the highest quartile had high ovarian androgen levels and were leaner. These findings pointed to the ovaries in the etiology of their PCOS. In PCOS, increased testosterone production was related to inappropriate gonadotropin secretion almost fifty years ago [36,37]. An elevated LH/FSH ratio was used as the criterion to diagnose PCOS before the availability of androgen and AMH assays and ultrasound. Currently, the demonstration of elevated LH and FSH levels is not required for a PCOS diagnosis. Accordingly, 55% of our subjects did not have LH or FSH measurements in their records. The limited available data showed that the highest AMH quartile had higher LH levels as compared to quartile 2 but not quartiles 1 or 3. A possible reason for not seeing a distinct elevation in LH levels in the highest AMH quartile may be that their high serum testosterone levels may have partially suppressed their LH concentrations through a feedback mechanism. 

Even though our findings support the measurement of serum AMH in PCOS, we cannot conclude that the inclusion of serum AMH measurements adds value in the determination of the ovarian morphology by ultrasound, because our study did not include ultrasound data. However, these two methods provide different types of information, namely structural vs. biochemical. It is not known whether all women with PCOS secrete the same amount of AMH per cyst, but it is unlikely. If some cysts secrete larger amounts of AMH and AMH causes resistance to FSH in the follicle, it is conceivable that even if two women have the same number of ovarian cysts, the woman with the higher serum AMH level may have more reproductive problems than the one with the lower serum AMH level. This possibility needs to be addressed in future research. 

In conclusion, clinical heterogeneity in PCOS creates challenges in the diagnosis and management of this disorder. It is conceivable that women with varying serum AMH levels also have varying degrees of ovarian pathology contributing to their PCOS. Thus, they differ in their clinical and biochemical presentations and management needs. Our observations indicate that AMH-based phenotyping may be a step towards understanding the heterogeneity in PCOS and providing patient-centered care.

## Figures and Tables

**Figure 1 diagnostics-13-00500-f001:**
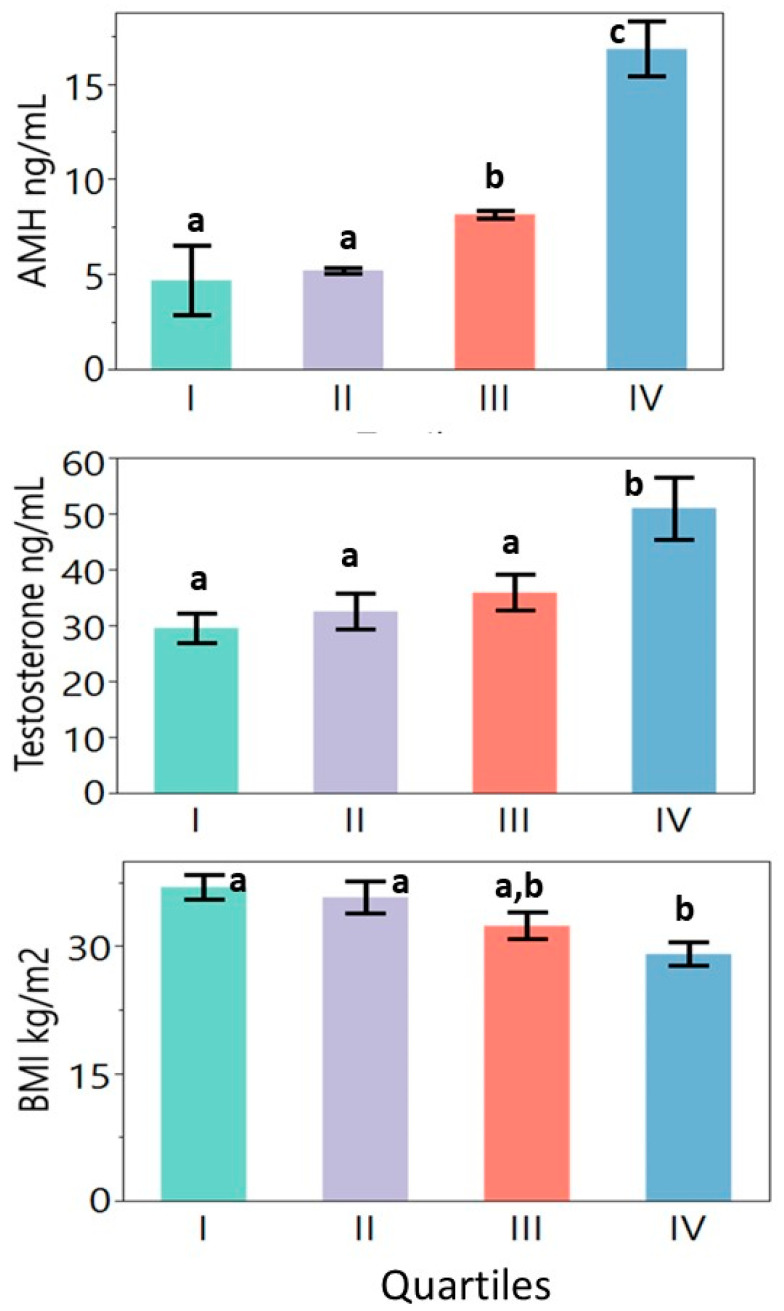
Differences in anti-Mullerian hormone, total testosterone, and body mass index values between quartiles. Continuous variables were evaluated using a one-way ANOVA followed by Tukey’s multiple comparison tests where relevant. Note: a, b, and c notations indicate significant differences among the quartiles at *p* < 0.05 (quartiles marked with the same letter do not differ significantly from each other).

**Table 1 diagnostics-13-00500-t001:** Clinical and biochemical variables of the women with PCOS.

Variables		N	Mean ± SD	Min	Max	Normal Reference Range for Age-Matched Females
**Clinical**	**Age (years)**	108	28.9 ± 6.3	17	45	
	**Weight (kg)**	106	91.5 ± 26.9	41	215.8	
	**BMI (kg/m^2^)**	106	33.7 ± 8.6	17.8	70.4	18.5 to 24.9 kg/m^2^
	**SBP (mmHg)**	104	118.1 ±1.0	92	147	
	**DBP (mmHg)**	104	77.1 ±0.1	55	107	
**Endocrine**	**AMH (ng/mL)**	108	8.5 ± 7.9	0.1	52.5	0.9–9.5 ng/mL
	**T (ng/dL)**	107	36.9 ± 20.6	6	151	8–60 ng/dL
	**SHBG (nmol/L)**	107	44.2 ± 31.2	6	185	2.2–14.6 pg/mL
	**Bioavailable T (ng/dL)**	106	16.7 ± 11.5	2.4	92.6	0.8–10.0 ng/dL
	**Free T (ng/dL)**	107	5.8 ± 4.1	1	32.5	0.3–1.9 ng/dL
	**DHEAS (ng/dL)**	102	229.8 ± 127.4	32	577	18–332 pg/dL
	**17OHP (ng/dL)**	67	59.9 ± 69.0	1	190	80 ng/dL
	**Prolactin (ng/mL)**	79	11.5 ± 6.1	3	31.8	4–30 ng/mL
	**LH (IU/L)**	49	7.3 ± 8.9	0.2	64	1.0–18.0 IU/mL
	**FSH (IU/L)**	49	4.9 ±1.9	0.3	9.9	2.0–12.0 mIU/mL
	**TSH (mIU/mL)**	102	1.8 ±0.8	0.75	6.3	0.4–4.5 mIU/mL
**Metabolic**	**HgBA1c (%)**	95	5.5 ± 0.5	4.5	7.9	4.0–5.6%
	**TC (mg/dL)**	50	179.4 + 31.5	134	236	<200 mg/dL
	**HDL-C (mg/dL)**	51	47.8 ± 10.0	33	76	>60 mg/dL
	**LDL-C (mg/dL)**	49	111.7 ± 33.7	57	239	<100 mg/dL
	**TG (mg/dL)**	50	118.0 ± 69.9	37	377	<150 mg/dL
			**Frequency %**			
**Subjective Symptoms**	**Amenorrhea**	37	34.3			
	**Oligomenorrhea**	67	62.0			
	**Menorrhagia**	16	14.8			
	**Infertility**	15	13.9			
	**Acne**	29	26.9			
	**Facial Hair**	56	51.9			
	**Hirsutism**	33	30.6			
	**Hair Loss**	21	19.4			
	**Obesity**	45	41.7			
	**Weight Gain**	44	40.7			

**Table 2 diagnostics-13-00500-t002:** Biochemical and clinical characteristics of women with PCOS based on AMH quartiles (means ± STD). The number of data points in each cell is presented in parentheses (n).

	1st Quartile (n = 27)	2nd Quartile (n = 25)	3rd Quartile (n = 29)	4th Quartile (n = 27)	ANOVA *p*
**AMH (ng/mL)**	2.1 ± 1.1 (27)	5.1 ± −0.6 (25)	8.0 ± 1.1 (29)	18.5 ± 9.9 (27)	**<0.0001**
**Age (years)**	31.2 ± 6.6 (27)	28.8 ± 6.2 (25)	28.0 ± 6.2 (29)	27.5 ± 5.9 (27)	0.2173
**Weight (kg)**	101.8 ± 22.0 ^b^ (26)	94.0 ± 20.6 (24)	93.3 ± 36.1 (29)	77.4 ± 18.7 (27)	**0.0107**
**BMI (kg/m^2^)**	36.7 ± 7.2 ^b^ (26)	35.0 ± 7.6 ^a^ (24)	34.0 ± 10.6 (29)	29.4 ± 6.9 (27)	**0.0056**
**SBP (mmHg)**	117.8 ± 2.0 (27)	117.5 ± 0.6 (27)	116.9 ± 1.8 (27)	120.7 ± 2.5 (27)	**0.4806**
**DBP (mmHg)**	77.9 ± 1.6 (27)	75.6 ± 1.9 (24)	75.7 ± 1.5 (27)	79.4 ± 2.1 (27)	**0.3510**
**Testosterone (ng/dL)**	26.5 ± 10.4 ^b^ (27)	34.4 ± 17.0 ^b^ (24)	35.1 ± 16.3 ^a^ (29)	51.3 ± 27.2 (27)	**0.0012**
**Bioavailable T (ng/dL)**	12.2 ± 6.6 ^a^ (27)	15.9 ± 9.0 (24)	16.5 ± 8.7 (28)	22.1 ± 17.0 (27)	**0.0359**
**Free T (ng/dL)**	4.4 ± 2.3 (27)	5.7 ± 3.2 (24)	5.6 ± 3.1 (29)	7.7 ± 6.0 (27)	0.0845
**SHBG (nmol/L)**	42.9 ± 33.5 (27)	42.3 ± 36.1 (24)	47.2 ± 28.2 (29)	48.9 ± 28.6 (27)	0.9782
**17-OHP (nmol/L)**	44.7 ± 46.4 (17)	41.5 ± 24.3 (16)	41.0 ± 20.0 (18)	79.9 ± 31.1 (16)	0.0768
**DHEAS (ng/dL)**	216.2 ± 115.1 (26)	235.1 ± 156.8 (23)	252.2 ± 128.3 (27)	215.5 ± 112.2 (26)	0.5054
**Prolactin (ng/mL)**	10.1 ± 6.1 (16)	12.8 ± 7.1 (17)	12.7 ± 6.4 (25)	9.9 ± 4.4 (21)	0.2600
**LH (IU/L)**	7.3 ± 2.5 (7)	4.8 ± 3.0 ^a^ (13)	5.0 ± 3.1 (14)	7.9 ± 2.8 (14)	**0.0332**
**FSH (IU/L)**	4.5 ± 1.4 (7)	4.6 ± 2.9 (13)	4.6 ± 1.4 (14)	5.3 ± 1.0 (14)	0.5919
**TSH (mIU/mL)**	1.7 ± 0.8 (25)	1.7 ± 0.1 (21)	2.2 ± 0.2 (2.6)	1.7 ± 0.1 (26)	0.2643
**HgBA1c (%)**	5.6 ± 0.5 (24)	5.6 ± 0.4 (22)	5.6 ± 0.7 (25)	5.3 ± 0.4 (24)	0.4388
**TC (mg/dL)**	177.5 ± 33.7 (15)	175.1 ± 31.2 (10)	187 ± 28.6 (12)	117.5 ± 34.2 (13)	0.4530
**HDL-C (mg/dL)**	44.8 ± 5.8 (15)	47.2 ± 8.0 (10)	46.5 ± 10.4 (12)	53.2 ± 13.5 (13)	0.0845
**LDL-C (mg/dL)**	111.3 ± 31.9 (15)	103 ± 21.1 (10)	125.8 ± 45.1 (12)	105 ± 30.6 (12)	0.1651
**TG (mg/dL)**	109.9 ± 50.1 (15)	125.7 ± 100.9 (10)	120.9 ± 60.0 (12)	118.7 ± 77 (13)	0.4812
**Oligomenorrhea (%)**	55.6 ± 50.6 (27)	60.0 ± 50.0 (25)	58.6 ± 50.1 (29)	74.1 ± 44.7 (27)	0.6948
**Amenorrhea (%)**	25.9 ± 44.7 (27)	28.0 ± 45.8 (25)	31.0 ± 47.1 (29)	51.9 ± 50.9 (27)	0.2116
**Infertility (%)**	7.4 ± 26.7 (27)	16.0 ± 37.4 (25)	13.8 ± 35.1 (29)	18.5 ± 39.6 (27)	0.7383
**Hirsutism (%)**	62.9 ± 49.2 (27)	48.0 ± 51.0 (25)	48.3 ± 50.8 (29)	48.1 ± 50.9 (27)	0.1892
**Alopecia (%)**	22.2 ± 42.4 (27)	12.0 ± 33.2 (25)	17.2 ± 38.4 (29)	25.9 ± 44.7 (27)	0.6208
**Weight Gain (%)**	37.0 ± 49.2 (27)	48.0 ± 51.0 (25)	55.2 ± 50.6 (29)	25.9 ± 44.7 (27)	0.1256

The number of data points in each cell is presented in parentheses (n). Clinical, biochemical, and anthropometric variables (continuous data) were compared across quartiles using ANOVAs followed by Tukey’s pairwise comparisons when the ANOVA was significant. Percentages of categorical variables were compared across quartiles using chi square likelihood tests for independence. Note: ^a^: *p* < 0.05 as compared to quartile 4; ^b^: *p* < 0.01 as compared to quartile 4.

**Table 3 diagnostics-13-00500-t003:** Relationships among clinical, biochemical, and subjective variables of women with PCOS were assessed using Spearman’s correlation. Correlation coefficients (rho) for those reaching significance (*p* < 0.05) are presented.

	Age	BMI	T	SHBG	Bio–T	Free–T
**AMH (ng/mL)**	−0.2102 (0.029)	−0.3795 (<0.0001)	0.4164 (<0.0001)		0.3232 (0.0007)	0.2760 (0.004)
**BMI (kg/m^2^)**				−0.3927 (<0.0001)		
**Bio T (ng/dL)**			0.8466 (<0.0001)	−0.4321 (<0.0001)		
**Free T (ng/dL)**			0.8422 (<0.0001)	−0.4532 (<0.0001)	0.9918 (<0.0001)	
**DHEA-S (ng/dL)**		−0.2149 (0.032)				
**HgBA1c (%)**		0.2217 (0.033)		−0.2180 (0.035)		
**Obesity** (subjective)		0.5948 (0.0001)		−0.3063 (0.001)		
**Weight Gain** (subjective)		0.2745 (0.004)		−0.3322 (0.001)		

## Data Availability

The data presented in this study will be available on request from the corresponding author, depending on the approval from the IRB.

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
