# Peer review of "Anti-Mullerian Hormone-Based Phenotyping Identifies Subgroups of Women with Polycystic Ovary Syndrome with Differing Clinical and Biochemical Characteristics"

_diagnostics, 2023, doi:10.3390/diagnostics13030500_

Round 1

Reviewer 1 Report

Dear Authors,

The presented study tackles an issue of Anti-Mullerian Hormone-Based Phenotyping of Sub-groups of Women with Polycystic Ovary Syndrome with Differing Clinical and Biochemical Characteristics. I have read the article with a great interest. The study was conducted reliably with appropriate selection of tests. Overall, I think that this article should be published, however some issues require complementary information:

1.       Verse 108-109 -Do we have any information about thyroid status of included patients? Hypothyroidism should be also in exclusion criteria due to similar picture of the ovaries which mimics PCOS?

2.       I suggest including limitation of the study.

Author Response

  1. Verse 108-109 -Do we have any information about thyroid status of included patients? Hypothyroidism should be also in exclusion criteria due to similar picture of the ovaries which mimics PCOS?

Thank you for the suggestions.  We reviewed the Electronic Medical records again. Ninety-eight patients out of 108 had TSH values available. Mean ± SD for TSH was 1.8 ± 0.1 mIU/mL. Ninety-six patients were euthyroid. Two patients had TSH above 4.5 mIU/mL (4.57 and 6.33 mIU/mL, respectively).  Five patients, including the patient who had the TSH = 6.33 mIU/mL, were taking levothyroxine.  We revised the Table 1 to include the TSH values for the entire study population, and Table 3 for each quartile. Based on these data, hypothyroidism is not the cause of PCO morphology in our study.

  1. I suggest including limitation of the study.

Thank you for helping us to eliminate this limitation.

Reviewer 2 Report

This is a study, which investigates AMH in PCOS. The data shows that AMH is higher in phenotypes with lower BMI and higher testosterone.

I suggest including data on blood pressure, menstrual pattern and hirsutism to explore the potential association between AMH and clinical outcomes in more detail. Further to elaborate in more detail on what the potential clinical use of AMH could be in the context of PCOS - does it add any value compared to ovarian morphology by ultrasound?

I strongly suggest replacing "PCOS-women" with "women with PCOS" throughout the manuscript.

Author Response

This is a study, which investigates AMH in PCOS. The data shows that AMH is higher in phenotypes with lower BMI and higher testosterone.

I suggest including data on blood pressure, menstrual pattern and hirsutism to explore the potential association between AMH and clinical outcomes in more detail.

Response: We included systolic and diastolic blood pressures in the Tables 1 and 3, and the frequencies of oligomenorrhea, amenorrhea, infertility, hirsutism, alopecia and weight gain for each quartile in the Table 3.

Further to elaborate in more detail on what the potential clinical use of AMH could be in the context of PCOS - does it add any value compared to ovarian morphology by ultrasound?

Response: This is a difficult question to answer with our current knowledge. Clearly, state-of-the-art ultrasound is the preferred method of determining ovarian morphology. However, ultrasound and serum AMH provide different kinds of information: structural vs. biochemical/endocrine. This is relevant for other endocrine glands. For example, two patients with the similar thyroid ultrasound findings may have very different thyroid function test results, or two identical looking adrenal nodules may have different hormone secretion patterns. It is not known if all women with PCOS secrete the same amount of AMH per cyst. If some cysts secrete larger amounts of AMH and AMH causes resistance to FSH at the follicle, it is conceivable that even when two women have the same number of ovarian cysts on the ultrasound, the woman with the higher AMH may have more reproductive problems than the women with lower AMH.  In this case, measurement of AMH would provide additional value. We hope future research will address this issue.

We now addressed this question in the second to the last paragraph of the manuscript.

I strongly suggest replacing "PCOS-women" with "women with PCOS" throughout the manuscript.

Response: All “PCOS-women” were replaced with “women with PCOS” throughout the manuscript.

Reviewer 3 Report

In this manuscript Nguyen et al present their retrospective study on 108 women with PCOS. They advocate that phenotyping these patients according to AMH levels leads to classification in subgroups with distinct clinical and biochemical characteristics.

Overall, this is a well-written and easy to follow article. However, the association of high AMH levels with high androgen levels and with lean PCOS patients is not exactly new.

I have the following comments for the authors:

- Lines 126-127: “bioavailable testosterone and free testosterone (calculated from testosterone and SHBG values)”. Which formula was used for the calculation? If it was the one introduced by Vermeulen, than albumin is also need to be measured. Please clarify.

- Line 135: “Each quartile consisted of 25 to 29 PCOS-women”. Why did the authors construct quartiles with different number of participants? Why not include 27 patients in each quartile?

- Table 1: are the mean values of Testosterone and SHBG correct? The mean T seems rather low and the mean SHBG rather high. Please double-check.

- It would be very interesting if the authors could provide information regarding the ultrasound characteristics (% of patients with polycystic morphology) and the number of patients with A, B, C, and D PCOS phenotype in each AMH quartile. This would make it possible to see weather AMH levels are associated with PCO morphology and with specific phenotypes of the syndrome.

- Table 3: was there a significant difference in T levels between quartiles 1 &2 or 1&3? Please report.

- Figure 1: please be more specific as to what the markers a, b, and c indicate (differences between which quartiles?).

Author Response

In this manuscript Nguyen et al present their retrospective study on 108 women with PCOS. They advocate that phenotyping these patients according to AMH levels leads to classification in subgroups with distinct clinical and biochemical characteristics.

Overall, this is a well-written and easy to follow article. However, the association of high AMH levels with high androgen levels and with lean PCOS patients is not exactly new.

I have the following comments for the authors:

- Lines 126-127: “bioavailable testosterone and free testosterone (calculated from testosterone and SHBG values)”. Which formula was used for the calculation? If it was the one introduced by Vermeulen, than albumin is also need to be measured. Please clarify.

Response: Bioavailable testosterone (assay# 0070134) and free testosterone (assay# 0081096) results were reported by the ARUP laboratories (Salt Lake City, UT). Based on their information “the concentration of bioavailable and free testosterone is derived from mathematical expressions based on constants for the binding of testosterone to albumin and/or sex hormone binding globulin”.

- Line 135: “Each quartile consisted of 25 to 29 PCOS-women”. Why did the authors construct quartiles with different number of participants? Why not include 27 patients in each quartile?

Response: The quartiles were based on the concentration of AMH and not ‘n’. In other words, we performed ‘Quantile binning’, where we split up the data by counting the number of values up to p’th percentile of AMH, in our case 25th, 50th, 75th, and 100th percentiles. ‘Equal frequency binning’ will divide the total ‘n’ into groups with equal bins. This would not allow us to use AMH cut-off values as a benchmark for further analysis as effectively as the currently used approach.  

Quantile binning resulted in 27 patients in quartiles 1 and 4 (similar to Equal Frequency Binning).  It resulted in 25 patients in quartile 2, and 29 patients in quartile 3 because there was a cluster of similar AMH values at the n=27 cut off. If we chose Equal Frequency Binning, patients with the same AMH values would have ended up in two different quartiles.

We now clarified this in methodology.

- Table 1: are the mean values of Testosterone and SHBG correct? The mean T seems rather low and the mean SHBG rather high. Please double-check.

Response: We double checked the values.  They are accurate.

- It would be very interesting if the authors could provide information regarding the ultrasound characteristics (% of patients with polycystic morphology) and the number of patients with A, B, C, and D PCOS phenotype in each AMH quartile. This would make it possible to see weather AMH levels are associated with PCO morphology and with specific phenotypes of the syndrome.

Response: Unfortunately, we did not have ultrasound reports in several patients.  In addition, since the UC Davis Medical Center PCOS Program is a referral practice, some of the ultrasounds were obtained in outside institutions. Many of the reports did not follow the diagnostic criteria: they used descriptive statements such as “multiple small peripheral follicles” without mentioning the size or number of cysts or ovarian volume.

- Table 3: was there a significant difference in T levels between quartiles 1 &2 or 1&3? Please report.

Response: As seen in Figure 1, there were no significant differences in testosterone levels between quartiles either 1 and 2, or 1 and 3.  

- Figure 1: please be more specific as to what the markers a, b, and c indicate (differences between which quartiles?).

Response: Quartiles marked with the same letter are not significantly different than each other. If we take the example of the AMH panel, quartiles 1 and 2 are marked as “a”, so they are not different than each other. Quartile 3 is marked as “b” and quartile 4 as “c”. So, quartile 3 is significantly different that quartiles 1, 2 and 4. Similarly, quartile 4 is different than quartiles 1, 2 and 3. In the BMI panel, quartiles 1 and 2 are not significantly different.  Quartile 4 is different than 1 and 2, but not quartile 3. Quartile 3 is not significantly different than any of the other quartiles.

We clarified the figure heading as suggested.